# SMARAN: Closing the Generalization Gap with Performance Driven Optimization Method

## Abstract

Optimization methods have evolved significantly by introducing various learning rate scheduling techniques and adaptive learning strategies. Although these methods have achieved faster convergence, they often struggle to generalize well to unseen data compared to traditional approaches such as Stochastic Gradient Descent (SGD) with momentum. Adaptive methods such as Adam store each parameter's first and second moments of gradients, which can be memory-intensive. To address these challenges, we propose a novel SMARAN optimization method that adjusts the learning rate based on the model's performance, rather than the curvature of the objective function. This approach is particularly effective for minimizing stochastic loss functions, standard in deep learning models. Traditional gradient-based methods may get stuck in regions where the gradient vanishes, such as plateaus or local minima. Therefore, instead of only depending on the gradient, we use the model's performance to estimate the appropriate step size. We performed extensive experiments on standard vision benchmarks, and the generalization trends observed with SMARAN demonstrate compelling distinctions relative to adaptive and non-adaptive optimizers.

## 1 Introduction

A stochastic optimization problem is defined as

$$\min_{\mathbf{x} \in X} \mathbb{E}_\xi[f(\mathbf{x}, \xi)] \tag{1}$$

where $\xi$ is a random variable that introduces uncertainty in the objective function $f(\mathbf{x}, \xi)$ and $\mathbf{x}$ is the decision variable belonging to the feasible domain $X$. Standard solution methods for this form of optimization problems include gradient-based approaches such as SGD and its variants Lecun et al. (1998); Graves et al. (2013); Krizhevsky et al. (2012). However, gradient-based methods rely on the gradient direction for updating the parameters, but the gradient itself is affected by the stochastic nature of the function; hence, a negative gradient direction may not always be the best search direction. In nonconvex settings, relying solely on the gradient magnitude to identify the optimal point can be misleading. Flat regions, saddle points, and inflexion points exhibit the property of zero gradient. Finally, gradients provide local neighborhood information; hence, one may become stuck at a local optimum instead of searching for a global solution. We then encounter the issue of gradient explosion in steep regions. Additionally, gradient-based optimizers lack adaptability in step size based on landscape curvature, resulting in uniform step size scaling. In the literature, several variants have been proposed to address these drawbacks. SGD with momentum Polyak (1964) and Nesterov Nesterov (1983) overcome the first problem by aggregating past gradients to determine the current update direction. Aggregation reduces the effect of the stochasticity in the gradient. Adaptive methods overcome the uniform scaling of the gradient along all coordinate directions. AdaGrad Duchi et al. (2011) was the first algorithm in this line of research. AdaGrad used the historical sum of squared gradients to adjust the learning rate of individual parameters, resulting in faster learning. However, the accumulated squared gradients grow monotonically, causing the learning rate to shrink and leading to premature convergence. Later methods, such as RMSProp Tieleman (2012) and Adam Kingma & Ba (2015), overcome this difficulty using an exponential moving average (EMA) of gradients. Adam is the most prominent optimizer used among the adaptive optimizers. It uses

the EMA of the gradient (the first-moment estimate) for the update direction. It normalizes the learning rate with the EMA of the gradient square, the second-moment estimate. Although Adam-based methods have the advantage of faster convergence, storing first and second-moment estimates for each parameter becomes memory-intensive. Additionally, there is no clear evidence that Adam-based methods generally outperform SGD with momentum in terms of generalization. To overcome these drawbacks of previous methods, we introduce a novel optimization approach based on the objective function value rather than gradient dependence. Main contributions of the paper are:

- A novel optimizer SMARAN, which is based on the concept of EMA but uses the objective function value to adjust the learning rate instead of gradients, unlike Adam-based methods, and also includes adaptive regularization.
- Theoretical regret analysis of our objective function for both convex and nonconvex settings, and provide bounds on the learning rate.
- We experimentally compared our algorithm with state-of-the-art methods. Experimental results demonstrate that SMARAN outperformed other methods in terms of generalization ability for vision tasks.

The key motivation for using objective value instead of gradients is the high sensitivity of gradients to stochastic noise. Additionally, objective value provides a more global measure of optimization progress, even in regions where the gradient vanishes or explodes. Normalizing the learning rate with EMA of squared loss provides a smoother convergence. Also, loss-driven adaptation encourages updates that are guided by overall performance rather than noisy local curvature, thereby improving generalization compared to gradient-based approaches. In stochastic optimization, optimizers that approach the global minimum without fully converging are often preferred, as this behavior tends to yield better generalization. SMARAN's adaptive learning rate is designed to achieve this effect while simultaneously ensuring faster convergence, as shown in Fig. 1.

## 2 RELATED WORKS

Progression from manually scheduled updates to gradient-based adaptivity marks a significant advancement in optimization methods. Classical methods such as SGD and its momentum-augmented variants Polyak (1964), including Nesterov Accelerated Gradient (NAG) Nesterov (1983), laid the foundation for this development. These methods primarily focused on exploiting gradient direction and aggregating past values to reduce noise from stochastic updates. Nevertheless, they fail to adapt the learning rate based on the loss landscape, which results in oscillations near the optimal point.

Learning rate scheduling schemes were introduced based on training steps to overcome these limitations. While step decay Ge et al. (2019) reduces the learning rate at predefined intervals, cosine annealing Loshchilov & Hutter (2017) and cyclic schedules Smith (2017) use periodic changes. Although these methods improve convergence, they lack responsiveness towards the loss landscape and model performance. Their performance is heavily dependent on manual tuning of hyperparameters.

Recent works on adaptive learning rates modify the Polyak step size for stochastic nonconvex optimization Loizou et al. (2021b). Orvieto et al. (2022) demonstrates a polyak stepsize variant with decreasing stepsize that gives a convergence rate equivalent to gradient descent with proper initialization.

A paradigm shift occurred with the coming of adaptive methods such as AdaGrad Duchi et al. (2011), RMSProp Tieleman (2012), and Adam Kingma & Ba (2015). They introduced a parameter-wise adaptation of the learning rate. AdaGrad uses accumulated squared gradients to normalize the learning rate, penalizing frequently updated directions but often resulting in premature convergence. RMSProp overcomes this drawback by using the EMA of squared gradients, promoting smoother adaptation. Adam combines first and second-order moment estimates of gradients for learning rate adaptation. This results in stable updates with rapid initial convergence. Empirical studies show that adaptive methods may overfit, resulting in inferior performance to SGD with momentum on specific benchmarks Chen et al. (2020); Reddi et al. (2018).

More recent optimization approaches include AdamW Loshchilov & Hutter (2019), which decouples the weight decay from gradient updates. AMSGrad Reddi et al. (2018) controls the learning rate to become monotonically decreasing over iterations. Yogi Zaheer et al. (2018) prevents the

second moment estimate of Adam from exploding by sign correction. AdaBound Luo et al. (2019) clips the bounds of the learning rate to avoid the exploding and vanishing problem. PAdam Chen et al. (2020) scales the learning rate by a tunable adaptivity parameter. RAdam Liu et al. (2020) addresses the variance in adaptive learning rates. AdaBelief Zhuang et al. (2020) replaces the uncentred second moment with a centred variance estimate around the gradient. DecGD Shao et al. (2025) decomposes the gradient into a product of the surrogate loss and its gradient, which allows the learning rate adaptation based on the loss vector. Recent meta optimizers include L4 optimizer Rolinek & Martius (2018), which uses the difference in loss values for learning rate scheduling and can be applied over any optimizer.

Now that we have traversed the evolution of optimization methods from simple gradient heuristics to momentum-based adaptation, we introduce our novel optimizer, SMARAN, whose mechanism and theoretical insights are discussed in the next section.

## 3 METHODOLOGY

Let $f : \mathbb{R}^n \to \mathbb{R}$ denote the loss function to minimize and let $\mathbf{x} \in X \subseteq \mathbb{R}^n$ be an n-dimensional vector. We have three hyperparameters: the global learning rate $\eta$, the regularization coefficient $\lambda$, and the moment discount factor $\gamma$. We included the factor $\epsilon$ with a value $10^{-8}$ for the numerical stability of the division operations. The terms $\mathbf{m}_t$ and $v_t$ represent the moments of normalized gradient loss and square loss at iteration $t$, respectively, with initial values set to zero.

The previous section identifies three key factors that drive the design of optimization algorithms: update direction, update magnitude, and adaptiveness across different landscapes. Following these insights and the gaps in optimizing a stochastic function, we incorporated these aspects into our method, illustrated in Algorithm 1. The algorithm finds the optimal point using gradient information, momentum, a loss-based scaling mechanism for adaptivity, and variable regularization to avoid overfitting. The algorithm blends normalization, regularization, and an adaptive learning rate.

Following the Normalized Gradient Descent Shor (1985), to estimate the update direction, we use the normalized gradient $\hat{g}_t$ instead of the complete gradient $\nabla f(\mathbf{x}_t)$

$$\hat{\mathbf{g}}_t = \frac{\nabla f(\mathbf{x}_t).}{\|\nabla f(\mathbf{x}_t)\| + \epsilon} \tag{2}$$

Since the loss function is stochastic, we take the exponential average over past gradients for smoothing purposes to reduce the effect of uncertainty. The first moment is

$$\mathbf{m}_t = \gamma \mathbf{m}_{t-1} + \hat{\mathbf{g}}_t. \tag{3}$$

**Theorem 1.** *Let $f : \mathbb{R}^n \to \mathbb{R}$ be a differentiable loss function parameterized by $\mathbf{x}$, then the norm of the exponential average of the normalized gradients over some time step $t$, given by Eq. (3), is upper bounded as*

$$\|\mathbf{m}_t\| < \frac{1}{1 - \gamma} \tag{4}$$

*Proof.* Expanding Eq. (3) and using the triangle inequality,

$$\mathbf{m}_t = \sum_{\tau=1}^{t} \gamma^{t-\tau} \hat{\mathbf{g}}_\tau \implies \|\mathbf{m}_t\| \leq \sum_{\tau=1}^{t} |\gamma^{t-\tau}|.\|\hat{\mathbf{g}}_\tau\|$$

Since $\|\hat{\mathbf{g}}\| < 1$ due to $\epsilon$ factor in Eq. (2)

$$\|\mathbf{m}_t\| < \sum_{\tau=1}^{t} |\gamma^{t-\tau}| = \frac{1 - \gamma^t}{1 - \gamma} \tag{5}$$

As $t \to \infty$, sum of geometric progression becomes

$$\frac{1 - \gamma^t}{1 - \gamma} \to \frac{1}{1 - \gamma} > \|\mathbf{m}_t\|.$$

Moreover, the first moment is upper-bounded. Hence, this moment prevents gradient explosion, provided the multiplicative learning factor is not $\infty$. □

This approach is effective for high-dimensional or ill-conditioned landscapes.

Adapting the learning rate based on the curvature of the function is achieved using a performance-based factor, rather than relying on the gradients. Our performance-based factor is

$$\frac{f(\mathbf{x}_t)}{\sqrt{v_t} + \epsilon} \tag{6}$$

where

$$v_t = \gamma v_{t-1} + f(\mathbf{x}_t)^2 \tag{7}$$

**Theorem 2.** *Let $f : \mathbb{R}^n \to \mathbb{R}_+$ be a continuous loss function parameterized by $\mathbf{x}$, and $\{\mathbf{x}_t\}$ for $t = 1, 2, ..., T$ be a finite parameter sequence generated by gradient descent updates in $T$ iterations, then for every time step $t$,*

$$0 < \frac{f(\mathbf{x}_t)}{\sqrt{v_t} + \epsilon} < 1 \tag{8}$$

*where*

$$v_t = \sum_{\tau=1}^{t} \gamma^{t-\tau} f(\mathbf{x}_\tau)^2 \tag{9}$$

*Proof.* From Eq. (7), we have $v_t \geq f(\mathbf{x}_t)^2$ which implies,

$$\sqrt{v_t} \geq f(\mathbf{x}_t) \implies \sqrt{v_t} + \epsilon > f(\mathbf{x}_t) \implies \frac{1}{\sqrt{v_t} + \epsilon} < \frac{1}{f(\mathbf{x}_t)} \implies \frac{f(\mathbf{x}_t)}{\sqrt{v_t} + \epsilon} < 1$$

Since $f : \mathbb{R}^n \to \mathbb{R}_+$, both numerator and denominator are positive,

$$0 \leq \frac{f(\mathbf{x}_t)}{\sqrt{v_t} + \epsilon} < 1 \tag{10}$$

This factor is 0 only when $f(\mathbf{x}_t) = 0$. □

Previous methods Tieleman (2012); Kingma & Ba (2015); Reddi et al. (2018); Zhuang et al. (2020) use the EMA of the gradient square for normalizing the learning rate because gradients give the curvature of the landscape. However, in a nonconvex setting, the gradient magnitude changes rapidly, especially near steep curvature, resulting in an aggressive change in the learning rate, which causes the optimizer to converge slowly. Fig. 1e shows the trajectory for 100 steps of different optimizers on the Beale function, which is a nonconvex landscape. In our approach, the optimizer adjusts the learning rate based on its recent losses, resulting in less aggressive changes and, consequently, faster convergence. Unlike other methods that get stuck at local minima or landscapes where the gradient vanishes, our optimizer searches for paths to find the global minimum, particularly in regions with high loss values. For constant loss, like in flat regions, SMARAN's learning rate approaches $\sqrt{1 - \gamma}$ regardless of the loss magnitude (proof given in Appendix A.2). In an ideal case, as $\mathbf{x} \to \mathbf{x}^*$(global minima), $f(\mathbf{x}) \to 0$ hence,

$$\lim_{\mathbf{x}_t \to \mathbf{x}^*} \frac{f(\mathbf{x}_t)}{\sqrt{v_t} + \epsilon} = 0$$

To prevent overfitting, we incorporated a weight decay regularization term into the update step.

$$\mathbf{x}_{t+1} = \mathbf{x}_t - \eta \left( \frac{f(\mathbf{x}_t)}{\sqrt{v_t} + \epsilon} \right) (\mathbf{m}_t + \lambda \mathbf{x}_t) \tag{11}$$

AdamW inspires the weight decay, but unlike AdamW, which uses a constant weight decay, ours is an adaptive weight decay controlled by an adaptive learning rate term. Since the learning rate scheduler is based on the objective function value over training data, if the optimizer tries to overfit the training data, the same proportion of regularization prevents the model from overfitting.

Finally, SMARAN's memory requirement is lower than that of the adaptive methods, as the adaptive learning rate is a scalar quantity.

---

**Algorithm 1** SMARAN

---

**Input**: Initial vector $\mathbf{x}_0 \in X \subseteq \mathbb{R}^n$, loss function $f(\mathbf{x})$
**Parameter**: $\eta, \lambda, \gamma$
**Output**: $\mathbf{x}_T$
**Initialize**: $\mathbf{m_0} = \mathbf{0}$, $v_0 = 0$
1: **for** $t = 1$ **to** $T$ **do**
2:      $\hat{\mathbf{g}}_t = \frac{\nabla f(\mathbf{x}_t)}{\|\nabla f(\mathbf{x}_t)\| + \epsilon}$
3:      $\mathbf{m}_t = \gamma \mathbf{m}_{t-1} + \hat{\mathbf{g}}_t$
4:      $v_t = \gamma v_{t-1} + f(\mathbf{x}_t)^2$
5:      $\mathbf{x}_{t+1} = \mathbf{x}_t - \eta \left( \frac{f(\mathbf{x}_t)}{\sqrt{v_t} + \epsilon} \right) (\mathbf{m}_t + \lambda \mathbf{x}_t)$
6: **end for**
7: **return** $\mathbf{x}_T$

---

(a) SGD on convex landscape     (b) Adam on convex landscape     (c) SGDM on convex landscape

(d) SMARAN on convex land-scape

(e) Optimizers on non-convex landscape

Figure 1: Behaviour of optimizers on convex and non convex landscapes.

## 4 CONVERGENCE ANALYSIS

We perform a convergence analysis of the SMARAN algorithm and highlight a risk bound under a convex setting. First introduced in Duchi et al. (2011), convergence analysis in a convex setting was discussed in many of the later works on adaptive methods, including Adam Kingma & Ba (2015), AMSGrad Reddi et al. (2018), Adabound Luo et al. (2019), Adabelief Zhuang et al. (2020), and DecGD Shao et al. (2025).

### 4.1 ONLINE CONVEX OPTIMIZATION

Given the objective function $f_t : X \to \mathbb{R}$, the online convex optimization framework aims to minimize the regret $R(T)$

$$R(T) = \sum_{t=1}^{T} f_t(\mathbf{x}_t) - \min_{\mathbf{x} \in X} \sum_{t=1}^{T} f_t(\mathbf{x}) \tag{12}$$

Standard assumptions Duchi et al. (2011); Reddi et al. (2018); Hazan et al. (2016) of online convex optimization framework are as follows

**Assumption 1.** (1) The domain $X \subseteq \mathbb{R}^n$ is a bounded convex set; the diameter of $X$ is assumed bounded. For some bound $D$, $\|\mathbf{x} - \mathbf{y}\| \leq D \; \forall \; \mathbf{x}, \mathbf{y} \in X$. (2) $f_t$ is a convex function. (3) Gradient of $f_t, \nabla f_t$ is assumed to be bounded. For some bound $G$, $\|\nabla f_t\| \leq G, \forall \; \mathbf{x}_t \in X$.

**Theorem 3.** *Under Assumption 1, $\lambda \in (0,1)$, $\eta_t = \frac{\eta}{\sqrt{t}}$, $\eta > 0$, and $V_t = \frac{f(\mathbf{x}_t)}{\sqrt{v_t}+\epsilon}$, the regret bound of SMARAN is*

$$R(T) \leq \frac{GD^2\sqrt{T}}{\eta V} + G\lambda D^2 + G\eta(G+\lambda D)^2(2\sqrt{T}-1) \tag{13}$$

Proof of Theorem 3 is given in Appendix A.1. We conclude that, like previous adaptive methods Kingma & Ba (2015); Reddi et al. (2018); Luo et al. (2019); Zhuang et al. (2020); Shao et al. (2025), SMARAN also has an upper bound in $\mathcal{O}(\sqrt{T})$.

## 4.2 STOCHASTIC NON-CONVEX OPTIMIZATION

The standard assumptions for stochastic nonconvex optimization Chen et al. (2019) include

**Assumption 2.** (1) $f(\mathbf{x}_t)$ is lower bounded and differentiable i.e.,$\|\nabla f(\mathbf{x}) - \nabla f(\mathbf{y})\| \leq L\|\mathbf{x} - \mathbf{y}\| \ \forall \mathbf{x}, \mathbf{y}$ where $L$ is the Lipschitz constant. (2) The noisy gradient is unbiased and has independent noise. $\mathbf{g}_t = \nabla f(\mathbf{x}_t) + \xi_t$, $\mathbb{E}(\xi_t) = 0$, $\xi_t \perp\!\!\!\perp \xi_j \ \forall t, j, \ t \neq j$ (3) At step $t$, the algorithm can access a bounded noisy gradient, and the true gradient is also bounded. i.e., $\|\nabla f(\mathbf{x}_t)\| \leq G$, $\|\mathbf{g}_t\| \leq G$, $\forall t > 1$.

Based on the above assumptions, we have the following results

**Theorem 4.** *Under Assumption 2, $\gamma_t < \gamma \leq 1$, $\eta_t = \frac{\eta}{\sqrt{t}}$, $\eta > 0$, and $V_t = \frac{f(\mathbf{x}_t)}{\sqrt{v_t}+\epsilon} > c$, where $c$ is a constant, the expected gradient norm square is upper bounded as*

$$\min_{t\in[T]} \mathbb{E}(\|\nabla f(\mathbf{x}_t)\|^2) \leq \frac{L^2}{c\eta\sqrt{T}}\left(C_1\eta^2 G^2(1+logT) + 4C_2n\eta(\sqrt{T}-1) + 4C_3n^2\eta^2(1+logT) + C_4\right) \tag{14}$$

*where $C_1, C_2, C_3$ are constants independent of $T$ and $n$ and $C_4$ is independent of $T$.*

Proof of Theorem 4 is given in Appendix A.4.

## 5 EXPERIMENTAL RESULTS

We highlight the results of an extensive evaluation of the SMARAN algorithm on different benchmark datasets for the vision task. We empirically demonstrate the generalization capability of SMARAN over state-of-the-art models. All the experiments are performed on NVIDIA RTX A6000 GPU with Python 3.12.7 and Pytorch 2.6.0 + cu124. Code for the proposed optimizer is available here.

### 5.1 EXPERIMENTAL SETUP

We perform experiments on an image classification task over multiple datasets and models. We use the AR10 and CIFAR100 Krizhevsky & Hinton (2009) datasets, which comprise $60,000$ color images of resolution $32 \times 32$. CIFAR10 contains 10 object categories, and CIFAR100 includes 100 categories. We split the data into $50,000$ training samples and $10,000$ test samples. For the Tiny ImageNet dataset, we used $100,000$ training samples and $10,000$ test samples of size $64 \times 64$. Tiny ImageNet contains 200 unique categories.

The architectures used for image classification include ResNet50 He et al. (2015) and DenseNet121 Huang et al. (2017). The architectures follow the standard configurations available in the PyTorch package. We use SGD, SGD with momentum (SGDM), Adam, AdamW, RAdam, DecGD, and Prodigy Mishchenko & Defazio (2024) as optimizers for a comparative study with SMARAN for the image classification task. Unless otherwise stated, all optimizers are initialized with the default hyperparameter values as mentioned in the PyTorch official documentation. We use cross-entropy loss as an objective function. We train each optimizer with a list of learning rates in the range $[10^{-1}, 10^{-2}, 10^{-3}]$ to find the best-performing configurations. Fig. 2 compares different models with the best-performing configurations for the CIFAR10 and CIFAR100 datasets. SGD is configured with a learning rate of $0.1$, momentum of $0$, and weight decay. SGDM uses the same configuration with a momentum of $0.9$. Adam and AdamW follow a learning rate of $0.001$ with $\beta_1 = 0.9$,

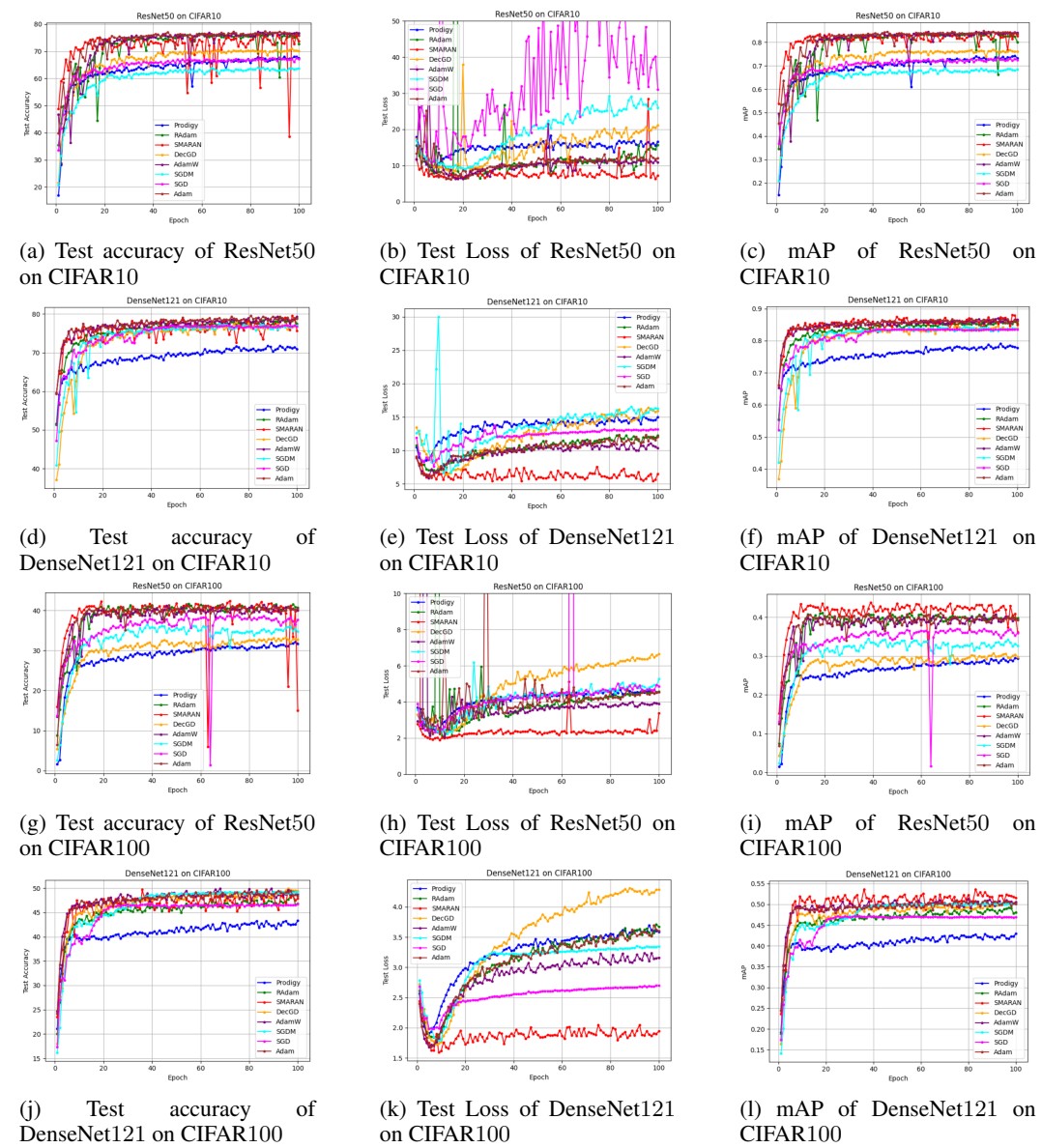

(a) Test accuracy of ResNet50 on CIFAR10

(b) Test Loss of ResNet50 on CIFAR10

(c) mAP of ResNet50 on CIFAR10

(d) Test accuracy of DenseNet121 on CIFAR10

(e) Test Loss of DenseNet121 on CIFAR10

(f) mAP of DenseNet121 on CIFAR10

(g) Test accuracy of ResNet50 on CIFAR100

(h) Test Loss of ResNet50 on CIFAR100

(i) mAP of ResNet50 on CIFAR100

(j) Test accuracy of DenseNet121 on CIFAR100

(k) Test Loss of DenseNet121 on CIFAR100

(l) mAP of DenseNet121 on CIFAR100

Figure 2: Experimental Results on CIFAR10 and CIFAR100 datasets.

$\beta_2 = 0.999$, $\epsilon = 10^{-8}$ and weight decay of 0 and 0.01 respectively. RAdam follows the same configuration as Adam. DecGD and Prodigy follow their respective default configurations Shao et al. (2025); Mishchenko & Defazio (2024). SMARAN uses a learning rate of 0.1, $\gamma = 0.9$, and $\lambda = 0.01$. We did not use any scheduling schemes, such as cosine annealing, for any of the optimizers mentioned above, as our objective is to compare the intrinsic performance of each optimizer with SMARAN. Therefore, any discrepancies between our results and the benchmark values reported in the literature can be attributed to either differences in the scheduling schemes or variations in the underlying architectures.

All models are trained for 100 epochs with a batch size of 128. According to their preprocessing schemes, only standard normalization is applied to the CIFAR10 and CIFAR100 images. For Tiny ImageNet, we normalize the data with a mean of $[0.480, 0.448, 0.398]$ and a standard deviation of $[0.277, 0.269, 0.282]$. We perform data augmentation by padding four pixels on all sides and cropping to a fixed resolution of $64 \times 64$. To augment orientation diversity, horizontal flipping is performed. All model weights are randomly initialized. We use three evaluation metrics for com-

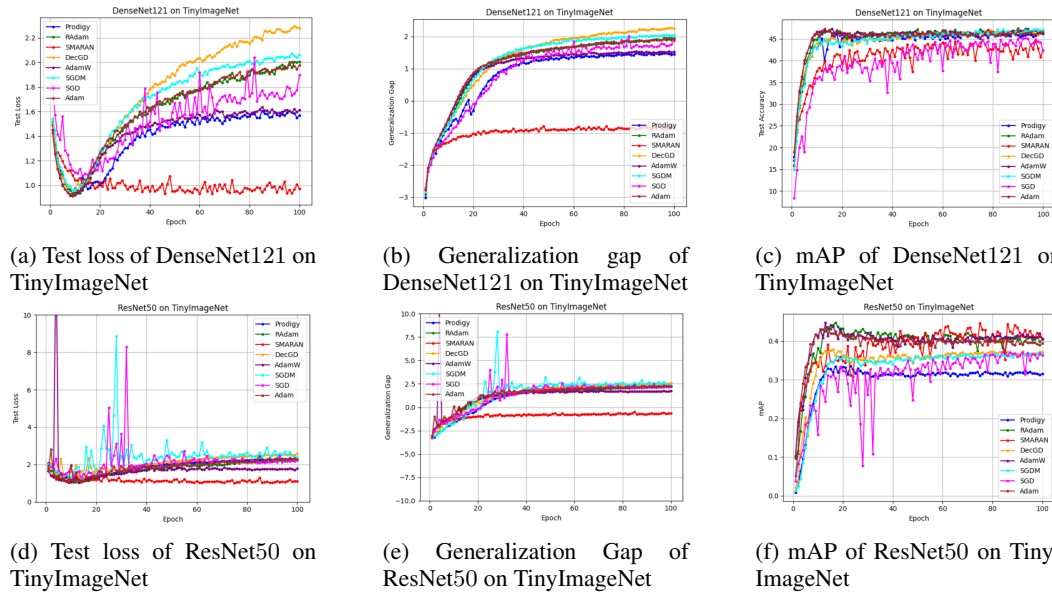

(a) Test loss of DenseNet121 on TinyImageNet

(b) Generalization gap of DenseNet121 on TinyImageNet

(c) mAP of DenseNet121 on TinyImageNet

(d) Test loss of ResNet50 on TinyImageNet

(e) Generalization Gap of ResNet50 on TinyImageNet

(f) mAP of ResNet50 on TinyImageNet

Figure 3: Experimental Results on Tiny Imagenet Dataset.

parison: test accuracy, test loss, and mean average precision (mAP) on the CIFAR10 and CIFAR100 datasets, as shown in Fig. 2. For the Tiny ImageNet data, we plot the generalization gap alongside test loss and mAP in Fig. 3. Apart from the optimizers mentioned above, we also experimented with other baselines that work based on the loss function, such as Stochastic Polyak Stepsize Loizou et al. (2021a), POlyak NOnmonotone Stochastic (PoNoS) Galli et al. (2023), and sign-based Lion Chen et al. (2023) optimizer. Results of the experiments are shown in Appendix A.3. Apart from the baseline optimizers, we compared our optimizer with L4 meta optimizer applied to Adam and SGDM, which Rolinek & Martius (2018) refers to as L4 Adam and L4 Mom in Fig. 4. The L4 meta optimizer significantly enhances the performance of Adam and SGDM, yielding generalization performance comparable to SMARAN. However, for DenseNet121, SMARAN outperforms both L4 Adam and L4 Mom in terms of test loss and generalization gap.

To analyze the sensitivity of hyperparameters on SMARAN's performance, we conducted a study to find the best possible configurations. We have used ResNet18 as the model and performed experiments on the CIFAR10 dataset. By systematically varying the values of these hyperparameters, we analyzed their effects on the loss value, as shown in Fig. 5. The best configuration values found are $\eta = 0.0436, \lambda = 0.0248$, and $\gamma = 0.8912$.

## 5.2 DISCUSSION

According to the results shown in Fig. 2, the SMARAN algorithm outperforms all other models in terms of test loss. Moreover, when comparing the test accuracy, SMARAN outperforms the state-of-the-art optimizers. SMARAN's training curve tends to stabilize in regions where overfitting is prevalent. Where other optimizers overfit the data after a certain number of epochs, SMARAN is a perfect fit on the data, with the testing loss either decreasing, as in the case of CIFAR10, or remaining stable. The reason behind this behaviour is the variable regularization parameter in our optimizer formulation. One can also see the same phenomenon for AdamW; however, for AdamW, the regularization is fixed, whereas for SMARAN, the adaptive learning rate parameter controls the regularization. The optimizer aims to minimize the gap between the training loss and the test loss, thereby improving generalization. Figs. 3b and 3e show the generalization gap, which is the difference between the testing and training losses of DenseNet121 and ResNet50 on the TinyImageNet dataset. The results show that the generalization gap is closer to zero for SMARAN compared to other optimizers. Generalization gap plots of other datasets are provided in the Appendix A.3. Since the accuracy curve does not reflect the proportionate improvement displayed by the loss curve, we use mAP as a complementary evaluation metric. The mAP values shown in Figs. 2 and 3 suggest

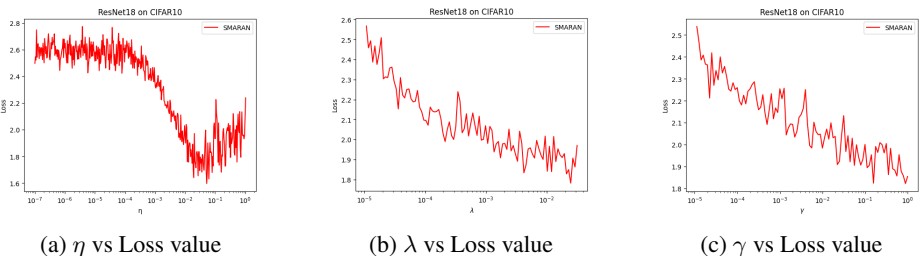

Figure 4: Comparison of SMARAN with L4 Adam and L4 Mom.

(a) $\eta$ vs Loss value

(b) $\lambda$ vs Loss value

(c) $\gamma$ vs Loss value

Figure 5: Hyperparamater sensitivity analysis of SMARAN on ResNet18 on CIFAR10.

that SMARAN performance is better than other methods. Figs. 2h and 2k show that SMARAN performs better with more categories such as CIFAR100 over CIFAR10. A similar trend is seen with the TinyImageNet dataset in Figs. 3a and 3d. The results of the experiments on loss-based optimizers indicate that SMARAN outperforms other optimizers. The performance gap is relatively low in terms of accuracy on DenseNet121 with CIFAR100 compared to SPS and PoNoS, indicating a scope for further improvement with larger models and diverse datasets. Similarly, comparison with L4 Adam and L4 Mom in Fig. 4 demonstrates that, although SMARAN has outperformed state-of-the-art adaptive optimizers, there is still scope for further improvement.

## 6 CONCLUSION

In this work, we introduce a novel optimization method, SMARAN, which adapts the learning rate based on the loss value rather than relying on gradients. Our method provides a bounded learning rate, resulting in stable training and better generalization. Our variable regularization mechanism prevents the model from overfitting after achieving optimal test results. Although the model is not coordinate-wise adaptive, like Adam and other adaptive methods, the learning rate still adapts to the curvature of the loss landscape by using the exponential average of historical losses. Additionally, SMARAN is memory-efficient compared to Adam-type methods, as its learning rate is a scalar. Experimental results demonstrate the algorithm's effectiveness for vision-based models over adaptive methods. Future work includes extending the SMARAN optimizer to other domains, such as text and video processing.

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

# A APPENDIX

## A.1 PROOF OF THEOREM 3

*Proof.* The potential function is defined as

$$\phi = \|\mathbf{x}_{t+1} - \mathbf{x}^*\|^2 \tag{15}$$

$$\begin{aligned}
\|\mathbf{x}_{t+1} - \mathbf{x}^*\| &= \|\mathbf{x}_t - \eta_t V_t(\mathbf{m}_t + \lambda \mathbf{x}_t) - \mathbf{x}^*\|^2 \\
&= \|\mathbf{x}_t - \mathbf{x}^*\|^2 + \eta_t^2 V_t^2 \|\mathbf{m}_t + \lambda \mathbf{x}_t\|^2 - \eta_t V_t \langle \mathbf{m}_t + \lambda \mathbf{x}_t, \mathbf{x}_t - \mathbf{x}^* \rangle \\
&= \|\mathbf{x}_t - \mathbf{x}^*\|^2 + \eta_t^2 V_t^2 \|\mathbf{m}_t + \lambda \mathbf{x}_t\|^2 - \eta_t V_t \langle \mathbf{m}_t, \mathbf{x}_t - \mathbf{x}^* \rangle - \eta_t V_t \lambda \langle \mathbf{x}_t, \mathbf{x}_t - \mathbf{x}^* \rangle
\end{aligned}$$

rearranging the terms,

$$\langle \mathbf{m}_t, \mathbf{x}_t - \mathbf{x}^* \rangle = \frac{\|\mathbf{x}_t - \mathbf{x}^*\|^2 - \|\mathbf{x}_{t+1} - \mathbf{x}^*\|^2 + \eta_t^2 V_t^2 \|\mathbf{m}_t + \lambda \mathbf{x}_t\|^2 - \eta_t V_t \lambda \langle \mathbf{x}_t, \mathbf{x}_t - \mathbf{x}^* \rangle}{\eta_t V_t}$$

substitute $\mathbf{m}_t$ from Eq. 3,

$$\langle \gamma \mathbf{m}_{t-1} + \hat{\mathbf{g}_t}, \mathbf{x}_t - \mathbf{x}^* \rangle = \frac{\|\mathbf{x}_t - \mathbf{x}^*\|^2 - \|\mathbf{x}_{t+1} - \mathbf{x}^*\|^2 + \eta_t^2 V_t^2 \|\mathbf{m}_t + \lambda \mathbf{x}_t\|^2 - \eta_t V_t \lambda \langle \mathbf{x}_t, \mathbf{x}_t - \mathbf{x}^* \rangle}{\eta_t V_t}$$

$$\langle \hat{\mathbf{g}_t}, \mathbf{x}_t - \mathbf{x}^* \rangle = \frac{\|\mathbf{x}_t - \mathbf{x}^*\|^2 - \|\mathbf{x}_{t+1} - \mathbf{x}^*\|^2 + \eta_t^2 V_t^2 \|\mathbf{m}_t + \lambda \mathbf{x}_t\|^2 - \eta_t V_t \lambda \langle \mathbf{x}_t, \mathbf{x}_t - \mathbf{x}^* \rangle}{\eta_t V_t}$$
$$- \gamma \langle \mathbf{m}_{t-1}, \mathbf{x}_t - \mathbf{x}^* \rangle$$

substitute $\hat{\mathbf{g}}_t$ from Eq. 2 and for simplicity assume $\epsilon \approx 0$,

$$\langle \nabla f_t(\mathbf{x}_t), \mathbf{x}_t - \mathbf{x}^* \rangle = \|\nabla f_t(\mathbf{x}_t)\| \left( \frac{\|\mathbf{x}_t - \mathbf{x}^*\|^2 - \|\mathbf{x}_{t+1} - \mathbf{x}^*\|^2}{\eta_t V_t} + \eta_t V_t \|\mathbf{m}_t + \lambda \mathbf{x}_t\|^2 - \right.$$
$$\left. \lambda \langle \mathbf{x}_t, \mathbf{x}_t - \mathbf{x}^* \rangle - \gamma \langle \mathbf{m}_{t-1}, \mathbf{x}_t - \mathbf{x}^* \rangle \right) \tag{16}$$

From Eq. 12, regret is defined as

$$R(T) = \sum_{t=1}^{T} f_t(\mathbf{x}_t) - \min_{\mathbf{x} \in X} \sum_{t=1}^{T} f_t(\mathbf{x}) \tag{17}$$

$$= \sum_{t=1}^{T} f_t(\mathbf{x}_t) - \sum_{t=1}^{T} f_t(\mathbf{x}^*) \tag{18}$$

$$= \sum_{t=1}^{T} (f_t(\mathbf{x}_t) - f_t(\mathbf{x}^*)) \tag{19}$$

$$\leq \sum_{t=1}^{T} \langle \nabla f_t(\mathbf{x}_t), \mathbf{x}_t - \mathbf{x}^* \rangle \tag{20}$$

Therefore

$$R(T) \leq \|\nabla f_t(\mathbf{x}_t)\| \left( \frac{\|\mathbf{x}_t - \mathbf{x}^*\|^2 - \|\mathbf{x}_{t+1} - \mathbf{x}^*\|^2}{\eta_t V_t} + \eta_t V_t \|\mathbf{m}_t + \lambda \mathbf{x}_t\|^2 + \right.$$
$$\left. \lambda \langle \mathbf{x}_t, \mathbf{x}_t - \mathbf{x}^* \rangle - \gamma \langle \mathbf{m}_{t-1}, \mathbf{x}_t - \mathbf{x}^* \rangle \right) \tag{21}$$

Simplifying each term in Eq. 21 and substitute $\eta_t = \eta/\sqrt{t}$.

$$\sum_{t=1}^{T} \|\nabla f_t(\mathbf{x}_t)\| \left( \frac{\|\mathbf{x}_t - \mathbf{x}^*\|^2 - \|\mathbf{x}_{t+1} - \mathbf{x}^*\|^2}{\eta_t V_t} \right) \tag{22}$$

$$= \sum_{t=1}^{T} \|\nabla f_t(\mathbf{x}_t)\| \sqrt{t} \left( \frac{\|\mathbf{x}_t - \mathbf{x}^*\|^2 - \|\mathbf{x}_{t+1} - \mathbf{x}^*\|^2}{\eta V_t} \right) \tag{23}$$

$$\leq \frac{G}{\eta V} \sum_{t=1}^{T} \sqrt{t} \left( \|\mathbf{x}_t - \mathbf{x}^*\|^2 - \|\mathbf{x}_{t+1} - \mathbf{x}^*\|^2 \right) \tag{24}$$

$$\leq \frac{G}{\eta V} \left( \|\mathbf{x}_1 - \mathbf{x}^*\|^2 + \sum_{t=2}^{T} \sqrt{t} \|\mathbf{x}_t - \mathbf{x}^*\|^2 - \sum_{t=2}^{T} \sqrt{t-1} \|\mathbf{x}_t - \mathbf{x}^*\|^2 \right) \tag{25}$$

$$= \frac{GD^2}{\eta V} \left( n + \sum_{t=2}^{T} (\sqrt{t} - \sqrt{t-1}) \right) \tag{26}$$

$$\leq \frac{GD^2 \sqrt{T}}{\eta V} \tag{27}$$

Similarly,

$$\sum_{t=1}^{T} \|\nabla f_t(\mathbf{x}_t)\| \left( \lambda \langle \mathbf{x}_t, \mathbf{x}_t - \mathbf{x}^* \rangle \right) \tag{28}$$

$$\leq G\lambda D^2 \tag{29}$$

$$\tag{30}$$

For the remaining part,

$$\sum_{t=1}^{T} \|\nabla f_t(\mathbf{x}_t)\| \left( \eta_t V_t \|\mathbf{m}_t + \lambda \mathbf{x}_t\|^2 - \gamma \langle \mathbf{m}_{t-1}, \mathbf{x}_t - \mathbf{x}^* \rangle \right) \tag{31}$$

$$\leq \sum_{t=1}^{T} \|\nabla f_t(\mathbf{x}_t)\| \left( \eta_t V_t \|\mathbf{m}_t + \lambda \mathbf{x}_t\|^2 \right) \tag{32}$$

$$\leq G\eta \sum_{t=1}^{T} \frac{\|\mathbf{m}_t + \lambda \mathbf{x}_t\|^2}{\sqrt{t}} \tag{33}$$

$$\leq G\eta (G + \lambda D)^2 \sum_{t=1}^{T} \frac{1}{\sqrt{t}} \tag{34}$$

$$\leq G\eta (G + \lambda D)^2 (2\sqrt{T} - 1) \tag{35}$$

From Eq. 21, 27, 30 and 35 we write

$$R(T) \leq \frac{GD^2 \sqrt{T}}{\eta V} + G\lambda D^2 + G\eta (G + \lambda D)^2 (2\sqrt{T} - 1) \tag{36}$$

$\square$

### A.2 PROOF OF CONSTANT LEARNING RATE FOR CONSTANT LOSS

Given

$$V_t = \frac{f(\mathbf{x}_t)}{\sqrt{v_t} + \epsilon} \tag{37}$$

Assume $f(\mathbf{x}_t) = k$ where $k$ is some constant, and for simplicity, assume $\epsilon \approx 0$. Then from Eq. 7,

$$v_t = \gamma v_{t-1} + k^2 \tag{38}$$

The solution for the above recurrence relation is

$$v_t = \frac{k^2}{1 - \gamma} \tag{39}$$

substituting into Eq. 37, we get

$$V_t = \frac{k}{\sqrt{\frac{k^2}{1-\gamma}}} \tag{40}$$

$$= \sqrt{1 - \gamma} \tag{41}$$

### A.3 ADDITIONAL RESULTS

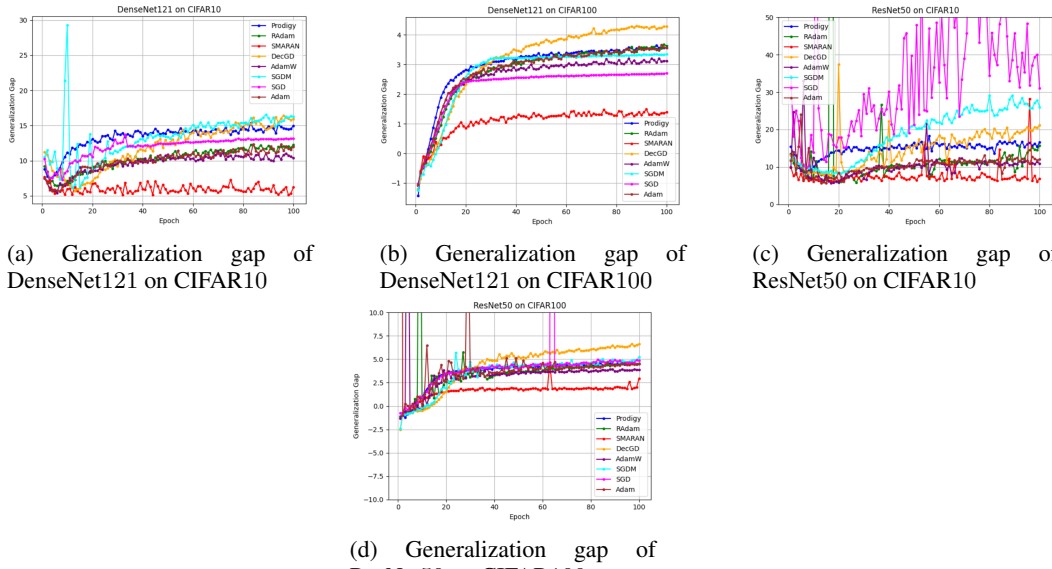

(a) Generalization gap of DenseNet121 on CIFAR10

(b) Generalization gap of DenseNet121 on CIFAR100

(c) Generalization gap of ResNet50 on CIFAR10

(d) Generalization gap of ResNet50 on CIFAR100

Figure 6: Generalization gaps on CIFAR10 and CIFAR100.

### A.4 PROOF OF THEOREM 4

Based on Assumption 2 and Chen et al. (2019), Eq. (3), we estimate

$$\mathbb{E}\left(\sum_{t=1}^{T} \eta_t \langle \nabla f(\mathbf{x}_t), V_t \nabla f(\mathbf{x}_t) \rangle \right) \leq \mathbb{E}\bigg( C_1 \sum_{t=1}^{T} \|V_t \eta_t \mathbf{g}_t\|^2 + C_2 n \sum_{t=2}^{T} |V_t \eta_t - V_{t-1} \eta_{t-1}|$$

$$+ C_3 n^2 \sum_{t=2}^{T-1} |V_t \eta_t - V_{t-1} \eta_{t-1}|^2 + C_4 \bigg) \tag{42}$$

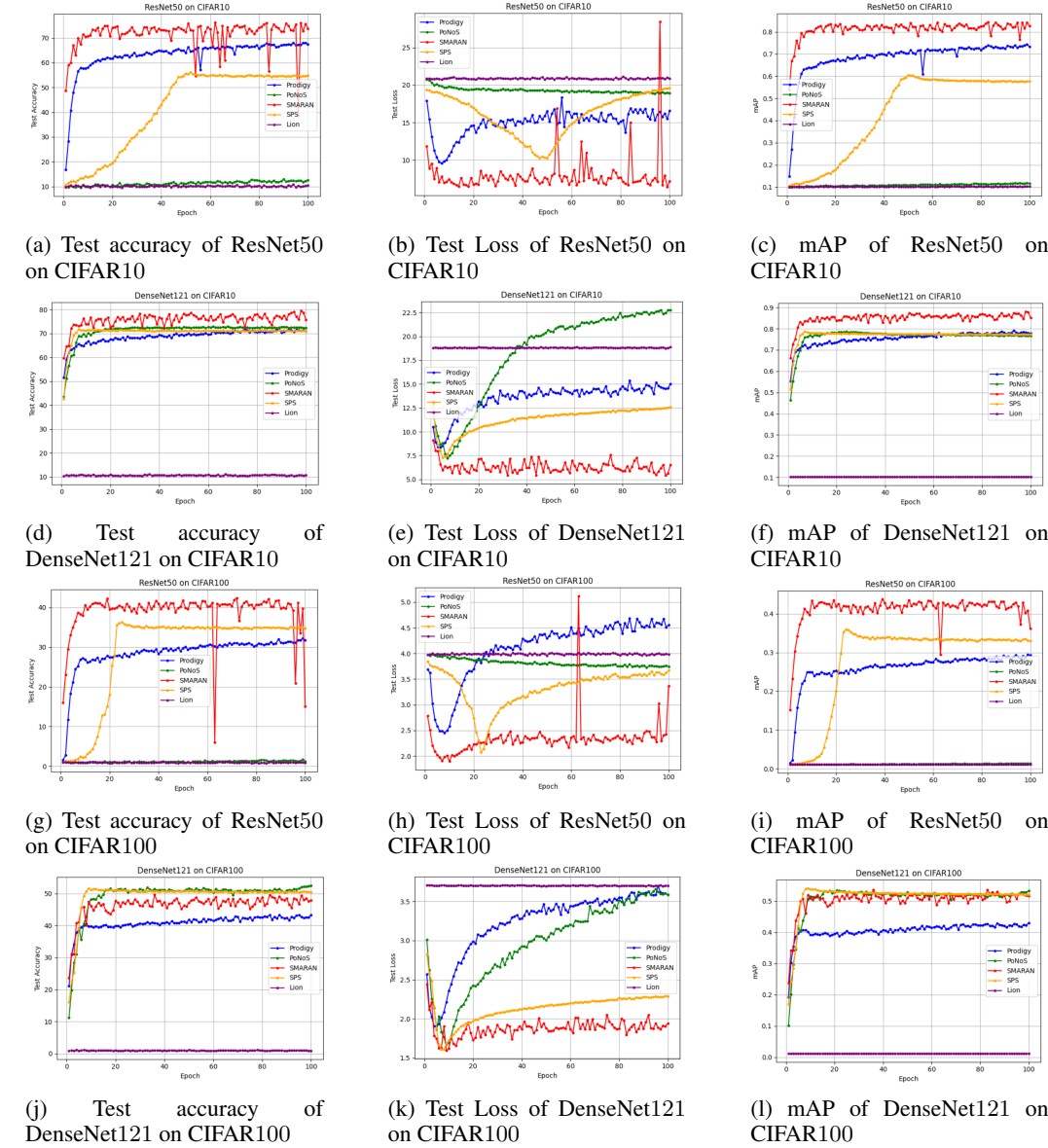

(a) Test accuracy of ResNet50 on CIFAR10

(b) Test Loss of ResNet50 on CIFAR10

(c) mAP of ResNet50 on CIFAR10

(d) Test accuracy of DenseNet121 on CIFAR10

(e) Test Loss of DenseNet121 on CIFAR10

(f) mAP of DenseNet121 on CIFAR10

(g) Test accuracy of ResNet50 on CIFAR100

(h) Test Loss of ResNet50 on CIFAR100

(i) mAP of ResNet50 on CIFAR100

(j) Test accuracy of DenseNet121 on CIFAR100

(k) Test Loss of DenseNet121 on CIFAR100

(l) mAP of DenseNet121 on CIFAR100

Figure 7: Experimental Results on CIFAR10 and CIFAR100 datasets for loss based models.

We have three expressions in the RHS of Eq. 42 to bound,

$$P_1 = \mathbb{E}\left( \sum_{t=1}^{T} \|V_t \eta_t g_t\|^2 \right) \tag{43}$$

$$P_2 = \mathbb{E}\left( \sum_{t=2}^{T} |V_t \eta_t - V_{t-1} \eta_{t-1}| \right) \tag{44}$$

$$P_3 = \mathbb{E}\left( \sum_{t=2}^{T-1} |V_t \eta_t - V_{t-1} \eta_{t-1}|^2 \right) \tag{45}$$

solving Eq. 43,

$$P_1 = \mathbb{E}\left(\sum_{t=1}^{T} \|V_t \eta_t \mathbf{g}_t\|^2\right) \tag{46}$$

$$\leq \mathbb{E}\left(\sum_{t=1}^{T} \|\eta_t \mathbf{g}_t\|^2\right) \tag{47}$$

$$\leq \eta^2 G^2 \mathbb{E}(\sum_{t=1}^{T} \frac{1}{t}) \tag{48}$$

$$\leq \eta^2 G^2 (1 + logT) \tag{49}$$

where the first inequality came from the fact that $V_t < 1$, Theorem 2, second inequality came from $g_t \leq G$ and the last inequality came from the fact $\sum_{t=1}^{T} \frac{1}{t} \leq 1 + logT$.

Solving Eq. 44 using the property $|a - b| \leq |a| + |b|$ and $V_t < 1$

$$P_2 = \mathbb{E}\left(\sum_{t=2}^{T} |V_t \eta_t - V_{t-1}\eta_{t-1}|\right) \tag{50}$$

$$\leq \sum_{t=2}^{T} \left(\frac{\eta}{\sqrt{t}} + \frac{\eta}{\sqrt{t-1}}\right) \tag{51}$$

$$\leq 2\eta \sum_{t=1}^{T-1} \frac{1}{\sqrt{t-1}} \tag{52}$$

$$\leq 4\eta(\sqrt{T} - 1) \tag{53}$$

Solving Eq. 45

$$P_3 = \mathbb{E}\left(\sum_{t=2}^{T-1} |V_t \eta_t - V_{t-1}\eta_{t-1}|^2\right) \tag{54}$$

$$\leq \sum_{t=2}^{T-1} (V_t \eta_t + V_{t-1}\eta_{t-1})^2 \tag{55}$$

$$\leq \sum_{t=2}^{T-1} 2(|V_t \eta_t|^2 + |V_{t-1}\eta_{t-1}|^2) \tag{56}$$

$$\leq \sum_{t=2}^{T-1} 2\left(\frac{\eta^2}{t} + \frac{\eta^2}{t-1}\right) \tag{57}$$

$$\leq 4\eta^2 \sum_{t=2}^{T-1} \left(\frac{1}{t-1}\right) \tag{58}$$

$$\leq 4\eta^2 (1 + logT) \tag{59}$$

So RHS of Eq. 42 becomes

$$RHS = C_1 \eta^2 G^2 (1 + logT) + 4C_2 n\eta(\sqrt{T} - 1) + 4C_3 n^2 \eta^2 (1 + logT) + C_4 \tag{60}$$

Now, let $V_t \geq c$, we have

$$V_t \eta_t \geq \frac{\eta c}{\sqrt{t}}$$

therefore the LHS of Eq. 42 becomes

$$\mathbb{E}\left(\sum_{t=1}^{T}\eta_t\langle\nabla f(\mathbf{x}_t), V_t\nabla f(\mathbf{x}_t)\rangle\right) \geq \mathbb{E}\sum_{t=1}^{T}\frac{\eta c}{\sqrt{t}}\|\nabla f(\mathbf{x}_t)\|^2 \tag{61}$$

$$\geq \frac{\eta c}{L^2}\mathbb{E}\sum_{t=1}^{T}\|\nabla f(\mathbf{x}_t)\|^2 \tag{62}$$

$$\geq \frac{\eta c}{L^2}\sqrt{T}\min_{t\in[T]}\mathbb{E}\|\nabla f(\mathbf{x}_t)\|^2 \tag{63}$$

Substituting the LHS and RHS from Eq. 60 and Eq. 63 to Eq. 42 yields

$$\min_{t\in[T]}\mathbb{E}(\|\nabla f(\mathbf{x}_t)\|^2) \leq \frac{L^2}{c\eta\sqrt{T}}\left(C_1\eta^2 G^2(1+logT) + 4C_2 n\eta(\sqrt{T}-1) + 4C_3 n^2\eta^2(1+logT) + C_4\right) \tag{64}$$

Hence proved.