# OpenReview forum: "SMARAN: Closing the Generalization Gap with Performance Driven Optimization Method"
_ICLR.cc/2026/Conference — Submitted to ICLR 2026_

### Official Review · Reviewer_bjWs · 2025-10-30

**Soundness:** 2
**Presentation:** 2
**Contribution:** 1
**Rating:** 2
**Confidence:** 4

**Summary:**

This paper proposes an optimization algorithm SMARAN for deep learning. SMARAN has two main characteristics, the first is that it normalizes the gradient before updating the first-order momentum, and the second is that it adopts the objective function value to update the second-order momentum. Then this work provides the analysis of the regret bound for SMARAN based on common assumptions. Finally, SMARAN is compared with several classical or adaptive optimizers in the experiments of CV tasks. SMARAN achieves great test accuracies on CIFAR datasets, and obviously a low generalization gap on Tiny-Imagenet.

The main contribution of this work is that it adopts the function value in the adaptive learning rate, which could reduce the memory cost of optimizer states.

**Strengths:**

This paper proposes an algorithm SMARAN which adopts the function value in the second-order momentum. This is a novel technique in the optimizer studies. The writing of this paper is clear. Most contents of this work are easy to understand.

**Weaknesses:**

The “Introduction” part lists a series of drawbacks of previous methods. However, the proposed method SMARAN seems not to overcome all these drawbacks, except for the large memory of Adam, which is not mentioned in the following parts. This work needs to emphasize the motivation for using the function value to calculate the learning rate in SMARAN. It is also not explained or discussed in the work why replacing the gradient with the function value could close the generalization gap.

Some statements in the work lack the support of references. For instance, it states that “Previous methods … because gradients give the curvature of the landscape. However, for a nonconvex setting, steep curvature results in slow learning, whereas in our approach”. I think some related works should be provided for these assertions.

The setting of the experiments is relatively simple, i.e. only conducting the CV tasks. SMARAN adopts an adaptive learning rate, and the common adaptive optimizers are good at training a transformer-based model. More experiments on this kind of model should be included.

The presentation of the experimental results needs to be improved. It would be better to summarize the specific values of the test accuracies in one list to make the results clearer. This work states that SMARAN is a memory-efficient optimizer. However, this point is not shown in the experiments.

In addition, the organization of this work is also poor. The formulas in the proof of theorems leave too many blanks in the paper.

**Questions:**

It is mentioned at the end of page 4 that “Since the learning rate scheduler is based on the objective function value over training data, if the optimizer tries to overfit the training data, the same proportion of regularization prevents the model from overfitting”. Could you give a more detailed explanation of why adopting the adaptive regularization factor, and what advantage it has over the constant regularization factor.

---

> ### Author Response · Authors · 2025-11-27
> **# Author Response to Reviewer bjWs**
>
> Thank you for your valuable feedback. We will revise the paper based on your feedback. Additional experimental results will be incorporated in the revised version of the paper.
>
> **Questions:**
> 1. It is mentioned at the end of page 4 that “Since the learning rate scheduler is based on the objective function value over training data, if the optimizer tries to overfit the training data, the same proportion of regularization prevents the model from overfitting”. Could you give a more detailed explanation of why adopting the adaptive regularization factor, and what advantage it has over the constant regularization factor.
>
> **Response:** Two key motivations were there for adopting adaptive regularization factors.
>
> Our objective is a stochastic function (loss function); hence, reducing the objective function to its optimal point will not guarantee generalization. We require an optimizer that will approach the neighborhood of the optimal solution but never reach it. Hence, we employed weight decay.
>
> Since regularization restrains the model weights from becoming very large, we require the restriction to be more stringent when the updates overshoot and less stringent when near the optimal. Moreover, loss value was a good measure of identifying the overshoot of weight parameters.
>
> Also, as mentioned in the reply to question 2 to Reviewer 4A8R, since regularization reduces the model's capacity to represent highly complex patterns, it prevents the model from learning noise (which we call overfitting of the data). One of the key limitations (if you could call it so) of adaptive weight decay is that it does not allow the model to reach the minimum point in the loss landscape. The model tends to approach the optimal point but never reaches it. The same is true for constant weight decay, but there, we can control how close we can reach. For stochastic optimization, reaching an optimum can lead to overfitting, as the training batch data defines the loss landscape. If one needs more generalization over a diverse dataset, use adaptive regularization instead of fixed weight decay.

---

### Official Review · Reviewer_7Jd3 · 2025-10-31

**Soundness:** 3
**Presentation:** 3
**Contribution:** 3
**Rating:** 4
**Confidence:** 4

**Summary:**

The paper proposes SMARAN, a novel optimization method for deep learning that adjusts the learning rate based on the model's performance (i.e., the objective function value) rather than the gradient's curvature, aiming to close the generalization gap often seen in adaptive optimizers. Unlike Adam, which uses exponential moving averages (EMAs) of gradients, SMARAN uses the EMA of past loss values to scale the learning rate and incorporates a form of adaptive weight decay to prevent overfitting. Experiments on vision benchmarks (CIFAR, Tiny ImageNet) show that SMARAN achieves better generalization, with lower test loss and smaller generalization gaps, compared to state-of-the-art optimizers like Adam, AdamW, and SGD with momentum.

**Strengths:**

1. The core strength is its demonstrated ability to achieve superior generalization performance, as evidenced by consistently lower test loss and significantly smaller generalization gaps across multiple datasets and architectures compared to popular baselines.
2. The key innovation of using the EMA of the loss value (rather than gradients) to adapt the learning rate is conceptually distinct from most existing methods. This approach allows the optimizer to be cautious when losses are high and accelerate when losses are low and decreasing, potentially avoiding pitfalls like vanishing gradients.
3. SMARAN is more memory-efficient than adaptive methods like Adam because its adaptive learning rate component is a scalar (based on the overall loss) rather than a vector (requiring storage of per-parameter moment estimates).

**Weaknesses:**

1. The experiments are confined to image classification tasks on standard vision datasets (CIFAR, Tiny ImageNet). The paper lacks evaluation on more complex tasks (e.g., language modeling, object detection) or larger-scale datasets (e.g., ImageNet), making it difficult to assess the method's broader applicability and scalability.
2. While the paper provides a regret bound in the online convex setting, deep learning involves highly non-convex optimization. The analysis does not fully address the behavior of SMARAN in this more relevant non-convex landscape, which is the primary context for its use.
3. The learning rate adaptation depends directly on the absolute value of the loss. If the loss function has a very different scale (e.g., due to different architectures or tasks), the hyperparameters (especially the global learning rate η) might need significant retuning, potentially reducing its claimed "adaptive" advantage in practice. The paper uses a fixed γ=0.9 and λ=0.01, but a more thorough ablation study on these hyperparameters would strengthen the claims.

**Questions:**

How does SMARAN’s performance-driven learning rate adaptation theoretically behave in non-convex landscapes, particularly near saddle points or flat regions where the loss value may remain nearly constant for many iterations?

---

> ### Author Response · Authors · 2025-11-27
> **# Author Response to Reviewer 7Jd3**
>
> We sincerely thank the reviewer for their careful reading and constructive feedback. Below we address each point in detail.
>
> ---
> **Weaknesses:**
>
> ### 1. Evaluation on more complex tasks.
> **Reviewer Comment:** The experiments are confined to image classification tasks on standard vision datasets (CIFAR, Tiny ImageNet). The paper lacks evaluation on more complex tasks (e.g., language modeling, object detection) or larger-scale datasets (e.g., ImageNet), making it difficult to assess the method's broader applicability and scalability.
>
> **Response:** Thank you for pointing this out. We agree that evaluating our method on larger-scale datasets, such as ImageNet, and on more complex tasks (e.g., language modeling, object detection), would further demonstrate its broader applicability. However, due to the substantial computational resources and training time required for complete ImageNet experiments, we opted for Tiny ImageNet as a practical proxy that still provides greater diversity and scale than CIFAR. This allowed us to validate our approach under more challenging conditions within our resource constraints. We will clarify this rationale in the revised manuscript and emphasize that extending to larger datasets and tasks is an important direction for future work.
>
> ---
> ### 2. Non convex analysis.
> **Reviewer Comment:** While the paper provides a regret bound in the online convex setting, deep learning involves highly non-convex optimization. The analysis does not fully address the behavior of SMARAN in this more relevant non-convex landscape, which is the primary context for its use.
>
> **Response:** Thank you for your suggestion. We will add the non-convex convergence analysis as a subsection under Convergence Analysis (Section 4.2 and Appendix A.4.) in the revised version of the paper.
>
> ---
> ### 3. Significant retuning due to scale variation.
> **Reviewer Comment:** The learning rate adaptation depends directly on the absolute value of the loss. If the loss function has a very different scale (e.g., due to different architectures or tasks), the hyperparameters (especially the global learning rate η) might need significant retuning, potentially reducing its claimed "adaptive" advantage in practice. The paper uses a fixed γ=0.9 and λ=0.01, but a more thorough ablation study on these hyperparameters would strengthen the claims.
>
> **Response:** We appreciate your concern regarding the effect of the loss function scale on hyperparameter tuning. However, we note that the learning rate does not directly depend on the absolute value of the loss. Instead, it depends on the ratio of the loss to its EMA, which makes it insensitive to the overall scale.
>
> ---
> **Questions:**
> ### 1. SMARAN behaviour in non-convex landscapes
> **Reviewer Comment:** How does SMARAN’s performance-driven learning rate adaptation theoretically behave in non-convex landscapes, particularly near saddle points or flat regions where the loss value may remain nearly constant for many iterations?
>
> **Response:** In flat regions where the loss value is nearly constant, the effective learning rate also becomes constant, governed by the discount factor $\gamma$ used in the EMA (see Appendix A.2 for proof). Near saddle points, since the effective learning rate does not depend directly on the gradient, and because SMARAN incorporates momentum from past gradients, it is less likely to get stuck compared to SGD.

---

### Official Review · Reviewer_4A8R · 2025-10-31

**Soundness:** 3
**Presentation:** 3
**Contribution:** 3
**Rating:** 8
**Confidence:** 4

**Summary:**

This paper introduces a novel optimization method which the authors term as SMARAN, that aims to bridge the generalization gap often seen with adaptive optimizers and improve memory efficiency. SMARAN uniquely adjusts its learning rate based on the model's performance (loss values), utilizing exponential moving average (EMA) of normalized gradients to determine the update direction and an EMA of squared loss values to dynamically scale the learning rate.

The authors argue that such a strategy based on loss performance allows for cautious learning with high losses and accelerated convergence in the regime with low and decreasing losses, thus preventing stagnation in flat regions.

SMARAN also integrates an adaptive weight decay regularization, whose strength is tied to the performance-based learning rate, to address overfitting. Theoretical analysis provided along with extensive experiments on image classification problems using multiple model architecture types.

**Strengths:**

I find a few key strengths in this work:
- The most significant strength is SMARAN's innovative approach to adjust the learning rate based on the objective function value (model or loss performance) rather than solely gradient information. This represents a fresh and unique direction in optimizer design.
- Proposed method SMARAN directly tackles two critical issues in deep learning: the generalization gap associated with adaptive methods and their memory intensiveness due to per-parameter moment storage. SMARAN's scalar learning rate is a clear win for memory efficiency, and the empirical results also seem to strongly support improved generalization.
- The integration of an adaptive weight decay mechanism, which is dynamically controlled by the performance-based learning rate, is a clever design. This dynamic regularization is well-suited for mitigating overfitting as the model converges.
- Proposed method consistently shows compelling empirical superiority across diverse vision benchmarks (CIFAR-10, CIFAR-100, Tiny ImageNet) and architectures (ResNet50, DenseNet121) when compared to a comprehensive set of baselines, including SGD, Adam, AdamW, RAdam, DecGD, and Prodigy.

**Weaknesses:**

Some weaknesses in the paper still remain undressed at this point:
- The use of a normalized gradient (Equation 2) could introduce numerical instability if the gradient norm approaches zero even when a reasonable epsilon value is used.
- While the paper states it's "performance-driven," the "performance" metric is consistently defined as the loss function value. Although loss is a direct indicator, further discussion on whether other performance metrics (e.g., validation accuracy, task-specific metrics) could be effectively incorporated into the adaptive learning rate calculation would be interesting. Wonder if the results would change in any way if we had looked these alternative performance metrics?
- Would have been good to see some theoretical study or explanations for how method performs in the non-convex setting.
- While future work mentions extending to text and video, the current experiments are primarily on vision tasks., it would be good to obtain empirical insights on language models / tasks as well.

**Questions:**

Some qns for the authors:
- How robust is the proposed SMARAN method to extremely noisy gradients, particularly concerning the normalized gradient? Could situations arise where the gradient norm is very small but not precisely zero, leading to an amplified noisy direction?
- The adaptive weight decay is a key feature. Could the authors elaborate on scenarios where a fixed weight decay (like in AdamW) might still be preferred, or where SMARAN's adaptive weight decay might have limitations?
- Could the authors offer more insight into the specific cases where SMARAN showed the largest performance gains or where its generalization gap was most significantly reduced compared to other optimizers? Are there particular types of datasets or model complexities where SMARAN particularly shines?

---

> ### Author Response · Authors · 2025-11-27
> **# Author Response to Reviewer 4A8R**
>
> We sincerely thank the reviewer for their careful reading and constructive feedback. Below we address each point in detail.
>
> ---
> **Weaknesses:**
>
> ### 1. Use of a normalized gradient
> **Reviewer Comment:** The use of a normalized gradient (Equation 2) could introduce numerical instability if the gradient norm approaches zero even when a reasonable epsilon value is used.
>
> **Response:** We appreciate your concern regarding the instability caused by the normalized gradient approaching zero; however, empirical results suggest otherwise. If the gradient norm approaches zero near the global optimum (i.e., the loss also approaches zero), then empirical results show stagnation.
> We plotted trajectory for 100 steps of different optimizers on the Beale function, which is a non-convex landscape. These results will be added to the appendix A.5. in the revised paper. However, if the loss is high and the gradient approaches zero, i.e., the instability that occurs will be controlled by the regularizer component in the update formula (Equation 13).
>
>  $x_{t+1}=x_t - \eta(\frac{f(x_t)}{\sqrt(v_t)+\epsilon})(m_t + \lambda x_t)$
> ---
>
> ### 2. "Performance" metric is consistently defined as the loss function value.
> **Reviewer Comment:** While the paper states it's "performance-driven," the "performance" metric is consistently defined as the loss function value. Although loss is a direct indicator, further discussion on whether other performance metrics (e.g., validation accuracy, task-specific metrics) could be effectively incorporated into the adaptive learning rate calculation would be interesting. Wonder if the results would change in any way if we had looked these alternative performance metrics?
>
> **Response:** We appreciate the suggestion to explore alternative performance metrics beyond the loss function value. In this work, we chose the loss value because it is universally available across tasks and provides a direct measure of optimization progress. For training the model, we have used the training loss for performance-driven optimization, as validation data should be reserved for hyperparameter tuning only.  However, we agree that incorporating validation metrics, such as validation loss and accuracy, or task-specific metrics into the adaptive learning rate calculation could be an interesting extension.
>
> ---
>
> ### 3. Theoretical study in the non-convex setting.
> **Reviewer Comment:** Would have been good to see some theoretical study or explanations for how method performs in the non-convex setting.
>
> **Response:** Thank you for your suggestion. We will add the non-convex convergence analysis as a subsection under Convergence Analysis (Section 4.2 and Appendix A.4.) in the revised version of the paper.
>
> ---
>
> ### 4. Extending  work to text and video.
> **Reviewer Comment:** While future work mentions extending to text and video, the current experiments are primarily on vision tasks., it would be good to obtain empirical insights on language models / tasks as well.
>
> **Response:** Thank you for the suggestion. Our study is scoped to vision tasks, chosen for clarity and consistency. While the method is in principle applicable to text and video, we focus here on establishing a solid foundation in vision, leaving broader applications to future exploration by the community.
>
> ---

---

> > ### Author Response · Authors · 2025-11-27
> >
> > **Questions:**
> >
> > ### 1. Robustness of SMARAN in noisy gradients.
> > **Reviewer Comment:**  How robust is the proposed SMARAN method to extremely noisy gradients, particularly concerning the normalized gradient? Could situations arise where the gradient norm is very small but not precisely zero, leading to an amplified noisy direction?
> >
> > **Response:** Please refer to the answer to weakness 1.
> >
> > ---
> > ### 2. Adaptive weight decay limitations.
> > **Reviewer Comment:** The adaptive weight decay is a key feature. Could the authors elaborate on scenarios where a fixed weight decay (like in AdamW) might still be preferred, or where SMARAN's adaptive weight decay might have limitations?
> >
> > **Response:** Since regularization reduces the model's capacity to represent highly complex patterns, it prevents the model from learning noise (which we call overfitting of the data). One of the key limitations of adaptive weight decay is that it does not allow the model to reach the minimum point in the loss landscape. The model tends to approach the optimal point but never reaches it. The same is true for constant weight decay, but there, we can control how close we can reach. For stochastic optimization, reaching an optimum can lead to overfitting, as the training batch data defines the loss landscape. If one needs more generalization over a diverse dataset, use adaptive regularization instead of fixed weight decay.
> >
> > ----
> > ### 3. Specific cases where SMARAN shines.
> > **Reviewer Comment:** Could the authors offer more insight into the specific cases where SMARAN showed the largest performance gains or where its generalization gap was most significantly reduced compared to other optimizers? Are there particular types of datasets or model complexities where SMARAN particularly shines?
> >
> > **Response:** SMARAN works best when the dataset is diverse. Similar to our experiments, SMARAN performed best for CIFAR-100 and TinyImageNet, which contain diverse images with a greater number of categories. SMARAN’s performance improves with deep architectures, such as DenseNet121, compared to shallow architectures.

---

### Official Review · Reviewer_Jv5d · 2025-11-01

**Soundness:** 2
**Presentation:** 2
**Contribution:** 2
**Rating:** 2
**Confidence:** 3

**Summary:**

The paper proposes an optimizer that adaptively changes its effective learning rate based on the an EMA of losses, in addition to using a EMA on the gradient history. Modulation of the learning rate as a function of the loss, is their main contribution. They provide a regret analysis in the online convex optimization setting. Empirically, across standard vision benchmarks they report competitive convergence with stronger generalization than Adam-style methods and SGD, while avoiding per-parameter second-moment storage.

**Strengths:**

1) The paper is easy to follow and is presented in a clear manner.
2) They propose a loss-driven gain that avoids per-parameter second-moment buffers reduces memory and implementation complexity.

**Weaknesses:**

1) Missing reference and comparison to [1].

2) I am afraid the hyper parameters with which SGD experiments are run are not optimal. For instance in Fig 1(a) SGD on ResNet-50 does less than 70% test accuracy, however in this [2] github implementation's default values SGD achieves 93.5%. Providing some clarity on this would be helpful in gauging the proposed methods efficicacy.


[1] Rolinek, Michal, and Georg Martius. "L4: Practical loss-based stepsize adaptation for deep learning." Advances in neural information processing systems 31 (2018).

[2] https://github.com/kuangliu/pytorch-cifar

**Questions:**

In line 128, it is mentioned that when the recent losses are low the learning rate is increased leading to faster convergence. However, looking at the update equation 13,$$
x_{t+1} = x_t - \eta \left( \frac{f(x_t)}{\sqrt{v_t} + \varepsilon} \right)\,(m_t + \lambda x_t)
$$
looking at this equation, I am afraid with lower loss the learning rate is decreased and not increased as stated in the paper. Can you provide clarity on line 128.

---

> ### Author Response · Authors · 2025-11-27
> **# Author Response to Reviewer Jv5d**
>
> We sincerely thank the reviewer for their careful reading and constructive feedback. Below we address each point in detail.
>
> ---
> **Weaknesses:**
>
> ### 1. Missing Reference and Comparison to L4
> **Reviewer Comment:** Missing reference and comparison to [1].
>
> **Response:**  We acknowledge that our manuscript did not explicitly cite L4: Practical loss-based stepsize adaptation for deep learning [1]. We experimented with the L4 Adam and L4 Mom and found that L4 Mom yielded the best results over 100 epochs of training compared to all other optimizers under consideration, including SMARAN. One point to note is that L4-based learning rate adaptation requires more iterations to reach the optimal value compared to other models. That can be considered a limitation that SMARAN does not share. Also the generalization gap of SMARAN is still lower for majority cases(see table below). We have used the implementation given in [3] for the L4 optimizer. We have used the default hyperparameter values given in Algorithm 1 of [1]. In the revised version, we will include a detailed comparison with L4, highlighting that while both methods adapt learning rates based on loss values, SMARAN integrates loss-driven adaptation with an EMA of both losses and gradients, and couples this with adaptive regularization. This combination provides bounded updates and improved generalization, while avoiding per-parameter second-moment storage and achieving an equivalent convergence rate to adaptive gradient-based optimizers. We agree that situating our contribution more clearly in relation to L4 will strengthen the paper.
>
> ### **Table : Test Accuracy and Generalization Gap across Optimizers, Models, and Datasets**
>
> | Optimizer      | CIFAR-10 ResNet-50 Accuracy | CIFAR-10 ResNet-50 Gap | CIFAR-10 DenseNet121 Accuracy | CIFAR-10 DenseNet121 Gap |
> |----------------|------------------------------|------------------------|-------------------------------|--------------------------|
> | SMARAN         | 73.7                         | 6.91                   | 76.68                         | 6.23                     |
> | L4_Mom         | 83.2                         | 6.12                   | 80.45                         | 8.10                     |
> | L4_Adam        | 79.2                         | 9.46                   | 75.54                         | 10.50                    |
> | Adam           | 79.2                         | 9.46                   | 75.76                         | 12.00                    |
> | SGD (momentum) | 68.5                         | 26.02                  | 73.45                         | 16.26                    |
>
> ---
>
> ### 2. Hyperparameter Settings for SGD Baseline
> **Reviewer Comment:** I am afraid the hyper parameters with which SGD experiments are run are not optimal. For instance in Fig 1(a) SGD on ResNet-50 does less than 70% test accuracy, however in this [2] github implementation's default values SGD achieves 93.5%. Providing some clarity on this would be helpful in gauging the proposed methods efficicacy.
>
> **Response:**
> We appreciate the reviewer’s concern regarding the reported SGD performance. The discrepancy noted arises primarily from differences in the training setup. The implementation referenced by the reviewer [2] employs cosine annealing for the learning rate schedule, which significantly boosts performance. When we ran their code without cosine annealing, the results were:
>
> Test accuracy: 71.11%
>
> Train accuracy: 77.05%
>
> Furthermore, their ResNet model is defined manually, whereas we used the PyTorch package implementation. When we replaced their model with the PyTorch-defined architecture (and removed cosine annealing), the results were:
> For SGD with momentum and weight decay:
>
> Test accuracy: 68%
>
> Train accuracy: 75%
>
> For SGD without momentum:
>
> Test accuracy: 77.79%
>
> Train accuracy: 94.53%
>
> Thus, the lower SGD baseline in our experiments reflects differences in scheduler choice and model definition, rather than a misconfiguration of the model. In the revised manuscript, we will clarify this point explicitly to avoid confusion and ensure fair comparison.
>
>
> [1] Rolinek, Michal, and Georg Martius. "L4: Practical loss-based stepsize adaptation for deep learning." Advances in neural information processing systems 31 (2018).
>
> [2] https://github.com/kuangliu/pytorch-cifar
>
> [3] https://github.com/iovdin/l4-pytorch/tree/master

---

> > ### Author Response · Authors · 2025-11-27
> >
> > ---
> >   **Questions:**
> > ### 1. Clarification on Line 128
> > **Reviewer Comment:** In line 128, it is mentioned that when the recent losses are low the learning rate is increased leading to faster convergence. However, looking at the update equation 13,
> >
> >  $x_{t+1}=x_t - \eta(\frac{f(x_t)}{\sqrt(v_t)+\epsilon})(m_t + \lambda x_t)$
> >
> > looking at this equation, I am afraid with lower loss the learning rate is decreased and not increased as stated in the paper. Can you provide clarity on line 128.
> >
> > **Response:** We agree with the reviewer’s observation that both the learning rate and the loss decrease over time. To investigate this, we simulated SMARAN’s behaviour on a convex function $z =(x-2) ^2 + (y+1)^2$.
> >
> > The plots confirm that when recent losses are consistently low and decreasing, both the effective learning rate and the step size decrease. This aligns with the reviewer’s interpretation, and we will revise the text to reflect this more accurately, ensuring that the explanation aligns with the update equation and empirical behavior.

---

### Author Response · Authors · 2025-12-04
**Final Rebuttal Comment**

We sincerely thank all reviewers for their thoughtful and constructive feedback. Both the positive and critical comments have been extremely valuable in helping us identify weaknesses, clarify our contributions, and strengthen the manuscript. We greatly appreciate the time and effort each reviewer has invested in carefully reading our work and providing detailed suggestions for improvement.

In the revised version, we have addressed all reviewer comments and highlighted our responses using a color‑coded scheme for clarity:

---

### Reviewer Jv5d – Green
- L4 meta optimizer comparison (Fig. 4, lines 405–410)
- Non‑optimal SGD performance clarification (lines 368–372)
- Correction of the line: *“when the recent losses are low the learning rate is increased leading to faster convergence”* (Lines 195–200 in the revised manuscript)

---

### Reviewer 4A8R – Brown
- Non‑convex setting analysis of convergence (Added Section 4.2 and Appendix A.4.)
- Plot of optimizer behaviour in the non‑convex Beale function (Fig. 1e, lines 195–203)

---

### Reviewer 7Jd3 – Magenta
- Non‑convex setting analysis of convergence (Added Section 4.2 and Appendix A.4)
- Behaviour of SMARAN on non‑convex landscapes such as flat regions (Lines 195–203, Appendix A.2)
- Hyperparameter sensitivity analysis for SMARAN (Fig. 5, lines 411–416)

---

### Reviewer bjWs – Blue
- Added motivation in the introduction (lines 61–78)
- Added references to Line 193
- Removed spacing issues in proofs (lines 152, 185)

---

### Common Comments – Red
- Addressed shared concerns across reviewers
- Behaviour of SMARAN in non‑convex setting (lines 193–203)
- Non‑convex convergence analysis (Section 4.2 and Appendix A.4)

---

This system enables easy tracing of each rebuttal and the corresponding changes made to the manuscript. We believe the paper has improved substantially thanks to the reviewers’ input, and we are grateful for their contributions to this process.

---

### Meta-Review · Area_Chair_YNoN · 2025-12-20

**Summary:**

The submission proposes a method called SMARAN, which is a memory-efficient optimizer that adapts a scalar learning rate using exponential moving averages of the loss and gradients to improve generalization over standard adaptive methods.

The reviewers overall agree that the idea of loss-driven learning-rate adaptation is interesting and empirically promising, at least on vision benchmarks. Another strength of the work mentioned by the reviewers is the improved generalization gaps compared to Adam and SGD.

However, two reviewers rated the contribution and soundness as fair or poor, citing missing or weak theoretical justification in the non-convex setting, limited task diversity (mostly CIFAR/Tiny ImageNet), and insufficient discussion of some prior work (mostly missing comparison to L4). Concerns were raised about potentially unfair or suboptimal SGD baselines, loss-scale sensitivity, numerical issues from normalized gradients, and lack of explicit memory-efficiency measurements.

Many comments were addressed in the rebuttal, including the addition of experiments comparing to L4 (although there are no details regarding the tuning of hyper-parameters), clarifying SGD settings, promising non-convex analysis, but some limitations (e.g. scope of the experiments, theory depth) remain. Overall, the reception is mixed: some reviewers recommend rejection due to clarity, theory, and experimental scope, while one reviewer strongly support acceptance based on novelty, generalization benefits, but I doubt the positive reviewer is aware of the prior work (L4). I think the limitations of the work need to be addressed in a revision, and I'm therefore not able to recommend acceptance. I do think the work has merits and look forward to seeing an updated version.

**Reviewer Concerns:**

See my response avove, the rebuttal does address some concerns but not all.

**Reviewer Scores:**

I'm not sure I can assess this well. I think the reviewers wouldn't have changed their scores given that the author's responses are quite limited.

---

### Decision · Program_Chairs · 2026-01-26

Reject